# Effects of Situation-Based Flipped Learning and Gamification as Combined Methodologies in Psychiatric Nursing Education: A Quasi-Experimental Study

**DOI:** 10.3390/healthcare10040644

**Published:** 2022-03-30

**Authors:** Haeran Kim, Boyoung Kim

**Affiliations:** 1Department of Nursing, College of Medicine, Chosun University, Gwangju 61452, Korea; rahn00@chosun.ac.kr; 2College of Nursing, Chonnam National University, Gwangju 61469, Korea

**Keywords:** situation, flipped learning, gamification, psychiatric, nursing

## Abstract

In psychiatric nursing courses, students learn about patients with mental illnesses or types of distress that may not be common in their daily lives. Consequently, some students experience difficulties interacting with patients. Therefore, new learning models that depart from the traditional one-way learning methods, engage students in clinical settings, and increase their motivation are needed. Situation-based flipped learning combined with gamification can improve nursing students’ care for patients with mental health problems. A total of 102 nursing students from a university in G Metropolitan City, Korea were randomly and equally divided into experimental and control groups. The experimental group participated in an eight-week psychiatric nursing program that included situation-based flipped learning combined with gamification. The control group participated in the traditional team-based lectures. After the program, both groups’ learning attitudes, problem-solving abilities, and empathetic responses were measured. The experimental group showed improved problem-solving (F = 4.012, *p* = 0.048) and empathetic abilities (t = −2.014, *p =* 0.047) after participating in situation-based flipped learning combined with gamification. The program was effective in helping students empathize with their patients and improve their problem-solving skills. Future curricula should be developed to incorporate flipped learning to nurture the practical competencies required for medical professionals, particularly in psychiatric nursing education.

## 1. Introduction

### Background

New learning models that break away from traditional educational methods are being developed to reduce costs and help students acquire more in-depth knowledge. Nursing education has stopped producing passive learners using instruction-based knowledge transfer methods and observation-oriented clinical practices [1]. However, traditional one-way teaching methods are used because of the professional class content and the number of classes per semester [2]. Further, they have limitations in improving the competence of nurses caring for patients with various health problems [3].

Blended learning differs from traditional one-way learning methods in that it usually combines online educational materials and opportunities for interaction online with traditional place-based classroom methods [4]. One of the methods used in blended learning is flipped learning (FL)—also known as backward learning, inverted instruction, and reverse learning. It is a reverse instruction method in which students engage in online pre-class learning followed by in-person lectures and discussions with the professor [5]. FL is divided into pre-class preparation based on various online audiovisual materials prepared by the professor before class and in-depth problem-solving during class time through multiple methods, such as discussions, games, role-playing, and think–pair–share activities [6]. Such student-centered education promotes higher-level thinking, such as applying, analyzing, and evaluating concepts beyond the level of simply memorizing and understanding concepts [7]. 

Gamification, or adding game mechanisms to non-game contexts, is useful in helping students engage in the learning environment and increases their motivation [8,9]. Applying gamification to learning can induce interest in learning concepts people are not usually interested in and can be a strategy to influence and motivate people’s behavior in class [10,11]. The nature of interaction through gamification involves learning how to communicate smoothly with others and helping students develop personal capabilities that enable collaboration [12]. 

Psychiatric nursing, one of many nursing courses, combines theory and practice based on understanding various mental disorders. In general, psychiatric nursing students may experience difficulties due to social prejudice against abnormal behavior or mental illness. Consequently, practical experience in psychiatric nursing is low compared to that in other types of clinical practice [13]. The content studied in psychiatric nursing is related to mental disorders that are not rarly encountered in daily life; therefore, it can cause anxiety and fear in students and be challenging [14,15]. To help motivate nursing students, gamification elements can be introduced into psychiatric nursing courses. 

It is essential to understand and empathize with the difficulties patients experience, especially when caring for individuals with mental impairments [16], which requires expertise, and an understanding of nursing intervention approaches appropriate for different situations [17]. Situation-based FL helps students learn diverse content based on specific mental illness cases. In a situation-based FL classroom, students tackle and solve the problems of situation-oriented cases in the classroom based on pre-class preparation through online content [18]. This method can improve students’ problem-solving skills by increasing the opportunities for active interaction between teachers and students [19,20]. Role-play, an element of gamification, can significantly influence understanding and empathizing with the mentally disabled during specific situations [21]. Furthermore, learning from case studies can improve problem-solving abilities and nurture the competencies required for medical professionals. 

Situation-based FL combined with gamification can improve student collaboration by allowing students to learn individually and repeatedly based on specific cases, promoting the sharing of learning content among student teams, and helping students acquire a high level of knowledge [22,23]. Therefore, game-based education and learning provide advantages by using gamification elements [24]. Students who have experienced these elements are more intrinsically motivated to achieve their goals and want to learn voluntarily [21,25]. Specifically, gamification is a form of education that enhances and supports the effectiveness of FL. 

This study applied situation-based FL combined with gamification to psychiatric nursing education to determine how it affects nursing students’ learning attitude, problem-solving abilities, and empathetic abilities. The application of situation-based FL combined with gamification reflects the innovative future of education and can provide implications for changing learning paradigms and promoting innovative educational practices. 

## 2. Materials and Methods

### 2.1. Research Design and Subject 

This quasi-experimental study used a pre-test–post-test nonequivalent control group design to understand how applying situation-based FL with gamification to psychiatric nursing education affects nursing students’ learning attitude, problem-solving ability, and empathetic ability. 

Participants were third-year nursing students at a four-year university in G-city. To prevent the spread of the experimental effect, the experimental group participated in 2016, and the control group participated in 2015. Of the 92 students in the experimental group and 85 in the control group, 51 in each group agreed to participate and were included in the analysis. Participants who did not consent to this study also trained in psychiatric nursing of the situation-based FL and gamification as combined methodologies. However, students who did not volunteer to participate did not respond to the survey. The required sample size was 102 people (based on a significance level α = 0.05, two groups, an effect size of 0.50, and a power of 0.80 for an independent sample *t*-test using G*Power 3.1.4 (Dusseldorf, Germany). This study used the effect size interpretation thresholds proposed by Cohen [26] for the independent sample *t*-test. 

### 2.2. Development and Application of Psychiatric Nursing FL Module

This study aimed to enhance the ability of nursing students to understand the behavioral characteristics of mental illness and to apply integrated psychiatric intervention techniques for the prevention, treatment, and rehabilitation of mental disorders by converging situational learning-based gamification. The course was designed to match students’ individual characteristics with different communication skills and knowledge levels to prepare them to meet mentally challenged patients for the first time in clinical settings. Researchers with five years of clinical experience in psychiatric nursing and three years of educational experience developed five FL video modules on schizophrenia, bipolar disorder, anxiety disorder, personality disorder, and drug disorder (40 min each). 

The instructional system was based on Dick and Carey’s [27] analysis, design, development, implementation, and evaluation model. This model is a systematic approach that analyzes educational needs to find solutions; designs, develops, and implements curricula; and evaluates the results. A needs analysis was conducted via a survey of students’ preferred teaching methods and learning activities. Students’ academic achievement levels were determined based on the previous year’s academic performance. Various videos and lecture materials for each class were prepared based on previous studies and the literature on FL. The researchers conducted courses on FL teaching methods to develop instructional and teaching materials. 

The design phase involved preparing the syllabus, which included a weekly schedule and a list of learning activities divided into out-of-class and in-class activities. Students’ performance goals were prepared based on a weekly schedule. The development phase involved developing educational materials on psychiatric nursing for out-of-class activities, providing lectures, and collecting activity data for in-class activities. 

The implementation phase consisted of actual class progress (out-of-class activity: one hour; in-class activity: two hours). Five modules comprising 40 min MP4 video lectures and five quiz questions were uploaded to the learning management system for the out-of-class activity. The modules allowed the students to understand the behavioral characteristics of schizophrenia, bipolar disorder, anxiety disorder, and personality disorder in each class and to apply the prevention, treatment, and rehabilitation intervention techniques for each mental disease. For the out-of-class activity, students were asked to watch a video lecture from the H University E-learning system on the theory of the disease and participate in the class. The video presented knowledge about each disease through videos, images, and sound. In-class activity was operated with mini-lectures and team-based learning. The students faced challenges, competition, self-expression, achievements, and rewards during the course, which are gamification elements. 

For the evaluation phase, the students took the individual readiness assurance test (iRAT) before each class to review the content they had learned while preparing for each session. This was followed by the team readiness assurance test (tRAT), in which they formed teams to discuss, solve, and submit answers to a quiz composed of the same questions as those in the iRAT. After completing the tRAT, the instructor shared the correct answers and gave mini-lectures to explain what the students had not understood. The question presented for the in-class activity was, “What is the therapeutic response that a nurse can provide when the subject says he/she is experiencing delusions and auditory hallucinations?” The questions in these situational cases were to be solved through team-based cooperative learning and competition. Additionally, after each session, the students wrote in reflection journals as a post-class activity (Figure 1). 

The developed situation-based FL combined with gamification is described in detail as follows: FL and gamification were applied during seven weeks of schizophrenia, bipolar disorder, anxiety disorder, personality disorder, and drug disorder modules. In particular, the gamification elements of challenge, competition, scaffolding, achievement, and compensation were applied to all in-class, out-class, and deeper-class domains the students participated in. For example, in the schizophrenia disease module, the goal of the out-of-class activity was accurately presented. Moreover, three or four missions (e.g., character identification, character interview, role-play, etc.) that will arouse the will to challenge were provided to increase the learner’s desire for achievement. Patients, guardians, and familiar people appeared in the presented situational cases of the five disease modules. The students were asked to analyze the presented characters and create interview questions. Based on the interview contents, the roles of the patient and the nurse were assigned to each, and nursing intervention was applied during the role-play for about 10 min. Furthermore, badges were provided to the students in the main roles and the teams that performed the role-play scenarios and demonstrations well. To ensure that the students who participated in this study were compensated, they were provided immediate feedback on their progress or results or rewarded with a badge. 

After completing of these missions, students could earn a total of three achievements and reward badges based on their level of effort. Students were able to check their learning effects and progress, and the results were shared on the leader board to encourage users to learn voluntarily. To stimulate their competitive spirit, the leader board provided the ranking of each team and the records of the opponents.

The content validity of this module was assessed by six experts, including two professors of psychiatry, two nurses specializing in mental health, and two nurses working in the department of psychiatry. The CVI (content validity) was 0.80. Based on the experts’ opinions, the situation scenarios were modified for the students to role-play using therapeutic communication methods and empathize with patients with auditory hallucinations or delusions. 

### 2.3. Traditional Teaching Method 

The control group completed the same curriculum (course content, credits) as that completed by the experimental group but did not adopt the blended learning strategy. Instead, this group learned about major mental disorders in 15 classes (three hours a week) that adopted the traditional learning method. The instructor gave lectures using PowerPoint and related videos and assigned two problem-based learning assignments to solve critical thinking problems that may arise in cases involving mental disorders (schizophrenia and major depressive disorder). The control group’s academic achievement was measured through similar midterm and final exams. 

### 2.4. Research Tools

#### 2.4.1. Learning Attitude 

We used a tool developed by Hwang [28] to evaluate students’ attitudes, habits, beliefs, and motivations toward class. This tool uses a five-point Likert scale (1 = not at all, 5 = very much), and the higher the score, the better the learning attitude. In Hwang’s study [28], Cronbach’s α was 0.84, and in this study, Cronbach’s α was 0.70.

#### 2.4.2. Problem-Solving Ability

We used a tool developed by Lee et al. [29] to measure problem-solving ability. It consists of 45 items in five categories (problem clarification, cause analysis, alternative development, planning/implementation, and performance evaluation) and nine sub-categories: (1) problem recognition, (2) information collection, (3) analysis ability, (4) divergent thinking, (5) decision-making, (6) planning ability, (7) execution, (8) risk-taking, and (9) evaluation and feedback. The tool uses a five-point Likert scale (never, rarely, sometimes, often, always), and the higher the score, the higher the problem-solving ability. Cronbach’s α was 0.94 while developing this instrument [29], and Cronbach’s α was 0.92 in this study.

#### 2.4.3. Empathetic Ability

Empathetic ability was measured using a tool developed by Wakabayashi et al. [30] and validated by Yeo [31] using the Korean version of the Empathy Quotient-Short form. It comprises 11 items on a two-point scale to measure empathy’s cognitive, emotional, and social aspects. The higher the total score, the higher the empathetic ability. In a study by Yeo [31], Cronbach’s α was 0.88, and in this study, Cronbach’s α was 0.81.

#### 2.4.4. Ethical Consideration

This study was conducted with the corresponding author working at H University after obtaining ethical approval from the Institutional Review Board of H University (IRB No. 1041223-201506-HR-079-01). The students voluntarily participated in this study. They had opportunities to ask questions and were allowed to refuse to provide information. The researchers explained the purpose and method of the study and obtained written informed consent from the participants before conducting the surveys. The students were treated equally regardless of whether they participated in the study. Furthermore, they were guaranteed anonymity, and their data were not shared with anyone other than the researchers. The researchers explained that the data would be coded and kept for three years after the research concluded and then discarded according to school regulations. 

### 2.5. Data Analysis Method

The collected data were analyzed using SPSS/WIN 25.0. The participants’ general characteristics, learning attitudes, problem-solving abilities, and empathy were analyzed using real numbers, percentages, means, and standard deviations. Homogeneity between the experimental and control groups was tested using Fisher’s exact test, x^2^ test, and *t*-test. Additionally, the Kolmogorov–Smirnov test was used to analyze normality. The post-test pre-test values were analyzed using an independent sample t-test to determine the program’s effectiveness. Furthermore, the variables with non-homogeneous post-test values between the two groups were tested using an ANCOVA. 

## 3. Results

### 3.1. Subject Characteristics and Study Variables

There were no significant differences between the two groups concerning gender, school records, school life satisfaction, or school life stress. According to the homogeneity test results for the participants’ learning attitudes, problem-solving abilities, and empathetic abilities, the average learning attitude scores of the experimental and control groups were 50.1 and 55.0, respectively, showing a statistically significant difference (t = 4.62, *p* = 0.001). Regarding problem-solving ability, the average scores of the experimental and control groups were 149.1 and 157.1, respectively. Furthermore, they showed a statistically significant difference (t = 2.72, *p* = 0.008). For empathy, the average scores of the experimental and control group were 41.4 and 41.3, respectively, with no statistically significant difference (t = −0.05, *p* = 0.961). Thus, the two groups were homogeneous (Table 1). 

### 3.2. Validating the Program’s Effectiveness

Table 2 shows the results of validating the effects of psychiatric nursing education by applying situation-based FL with gamification on nursing students’ learning attitudes, problem-solving abilities, and empathetic abilities. There was a significant difference in learning attitudes between the experimental and control groups (t = −2.555, *p =* 0.012). However, there was no statistical significance when treating pre-learning attitude as a covariate (F = 0.486, *p* = 0.487). There were statistically significant differences between the experimental and control groups with regard to problem-solving (F = 4.012, *p* = 0.048) and empathetic abilities (t = −2.014, *p =* 0.047).

## 4. Discussion

This study investigated how situation-based FL combined with gamification affected the learning attitudes, problem-solving abilities, and empathetic abilities of nursing students taking psychiatric nursing courses. 

There were no significant differences in the students’ learning attitudes after situation-based FL with gamification classes when treating the pre-values as covariates. Similarly, Kim and Kim [17] and Matsumoto [32] found that nursing students’ learning attitudes did not change significantly after applying FL. Furthermore, there was no significant change in learning attitudes in the experimental group. Nevertheless, there was a statistically significant difference between the pre-and post-intervention values of the experimental and control groups. Regarding learning attitude, Bae [33] reported that FL classes had a positive effect on improving students’ learning attitude in learning scenarios, and Tan et al. [34] confirmed that teaching satisfaction and learning attitude were more highly correlated with FL than they were with traditional methods. These results indicate that FL requires more active participation and initiative than conventional learning methods that mainly rely on passive learning. However, a study by Kim et al. [17] on nursing students reported conflicting results; therefore, more research is needed on this regard. 

After participating in situation-based FL with gamification classes, the experimental group had an improved learning attitude. Multiple studies on teaching methods that combine FL and gamification have been conducted in recent years, and they have shown considerable potential as new learning methods [35,36]. In one study, FL combined with gamification motivated students and enhanced interaction between teachers and students [37]. Furthermore, the study by Sánchez et al. [11] reported that FL combined with gamification had a positive effect on motivation and interaction among students. In this study [11,37], when presenting the answers to the assignments in each module, each team had to provide new answers that did not overlap with those of other teams based on the elimination method. Applying this method naturally induced competition between students, and their rewards and grades were distinguished by distributing stickers that could be used for assignments. Specifically, the classes had gamification elements to provide immediate feedback on their performance in the form of rewards, such as competition and stickers. Such immediate feedback plays a significant role in encouraging students to participate actively in the learning process and is a crucial element of gamification-enhanced FL. 

The students’ problem-solving abilities showed a significant change after situation-based FL classes combined with gamification. These results are consistent with those of a study by Kim et al. [17], which found significant differences in students’ problem-solving abilities, including analytical thinking, decision-making, and feedback after taking FL classes. A systematic literature review of the effect of FL in nursing education also reported improvements in critical thinking and problem-solving skills [34]. 

In situation-based FL, students engage in pre-class learning activities and solve specific cases during class. Allowing students to learn about nursing interventions while playing the roles of patients and nurses from various perspectives is meaningful because it enables them to have three-dimensional and lively learning experiences instead of accumulating fragmented knowledge. FL also facilitated interactions between students completing the assignments for each module. According to Park [9], elements such as interaction, sustainability, interest, and growth experience are significant concepts in gamification, while rewards, level-up, feedback, and experience are already used in education. Specifically, interest and immersion are essential elements in modern education [9], and using them to increase problem-solving abilities through interaction in group activities increases interest in learning and sustains motivation. 

There was a significant increase in students’ empathetic abilities after completing situation-based FL classes combined with gamification. This change was a result of role-playing where students were the subject involved in specific situations. The participants could improve their empathy by experiencing difficulties caused by auditory hallucinations and delusions of mentally challenged patients and by acquiring and training coping strategies for these symptoms. Situation-based FL is similar to problem-based learning because it presents problems and helps students acquire knowledge and develop problem-solving skills [38]. However, it is different from problem-based learning due to the feedback received during role-play. In the present study, the students solved problems similar to clinical settings to learn. Moreover, they increased their understanding of patients by empathizing with their roles as nurses and patients in role-play based on clinical situations. Their level of understanding is likely to significantly broaden the scope of understanding and empathizing with patients in the future. Further, it can be linked to experience [39], an essential concept in gamification. Kim and Kim [40] also reported a significant increase in emotional intelligence (or empathetic ability) after participating in a communication empowerment program based on the situated learning theory. Situation-based FL can help students accumulate knowledge and empathize deeply with patients by solving cases that require practice, such as biological and psychiatric nursing. 

The situation-based FL strategy applied with gamification as used in this study can enhance the effectiveness of psychiatric nursing education. Especially, team activities that students can achieve on their own pre-class, and through which students can analyze and apply clinical situations in the classroom, can expand thinking that integrates practice and theory [2,15]. This aspect can be found in the fact that students who were performing one-way learning in traditional psychiatric nursing education tried to apply it to clinical practice by themselves using communication and collaboration before and after class through FL. In the future, a situation-based FL strategy is expected to be useful for psychiatric nursing education.

Despite this significance, this study had limitations. First, although the two groups completed the same curriculum, they took part in the study in different years. The students in the experimental group may not have received equal opportunities because of the gamification elements in the classes, which may have caused those that fell behind to give up quickly. Therefore, it is necessary to find alternatives to compensate for unequal opportunities. Additionally, FL requires active participation both inside and outside the classroom. Consequently, it requires more time and effort than what is required in traditional learning methods. Thus, there is a need to motivate students to participate in the learning process without falling behind. Finally, there are limitations to generalizing the results because this study was conducted with university students from a specific department in one area. Therefore, future research should study whether the effects of situation-based FL with gamification are practical among other groups.

## 5. Conclusions

This study investigated how adopting situation-based FL combined with gamification affects nursing students’ learning attitude, problem-solving ability, and empathetic ability in psychiatric nursing education. The results showed statistically significant differences in problem-solving and empathetic abilities after the students completed psychiatric nursing education using FL. Additionally, this study adopted IT-enabled active learning to induce FL, monitor individual feedback and learning activities, and enable students to learn through team activities. The program in this study was effective because of the adoption of elements such as peer instruction and problem-based learning during the sessions. These results can be used to improve the educational environment of nursing education, especially for psychiatric and mental health nursing.

## Figures and Tables

**Figure 1 healthcare-10-00644-f001:**
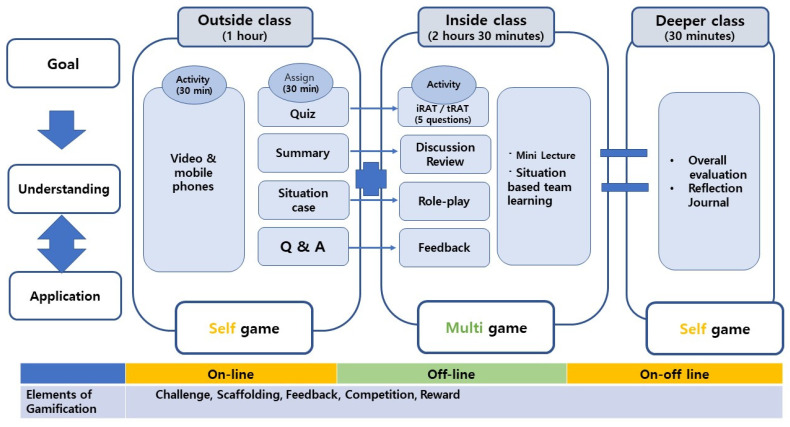
Situation-based flipped learning and gamification as combined methodologies in psychiatric nursing education. Abbreviations: FL: Flipped Learning; iRAT: Individual Readiness Assurance Test; tRAT: Team Readiness Assurance Test.

**Table 1 healthcare-10-00644-t001:** Participants’ characteristics and study variables (*N* = 102).

Characteristics		Exp. (*n* = 51)	Cont. (*n* = 51)	Total (*n* = 102)	x^2^ or t	*p*
		*N* (%) or M ± SD	*N* (%) or M ± SD	*N* (%)		
Gender	Male	9 (17.6)	8 (15.7)	17 (16.7)	0.071	0.790
	Female	42 (82.4)	34 (83.3)	85 (83.3)		
Academic achievement	High	22 (43.1)	21 (42.0)	43 (42.6)	0.823	0.663
	Moderate	20 (39.2)	23 (46.0)	43 (42.6)		
	Low	9 (17.6)	6 (12.0)	15 (14.9)		
Peer relationship satisfaction	High	6 (11.8)	7 (13.7)	13 (12.7)	2.166	0.339
	Moderate	34 (66.7)	27 (52.9)	61 (59.8)		
	Low	11 (21.6)	17 (33.3)	28 (27.5)		
Stress of college life	High	1 (2.0)	2 (3.9)	33 (42.6)	0.869	0.648
	Moderate	12 (24.0)	9 (17.6)	43 (42.6)		
	Low	37 (74.0)	40 (78.4)	15 (14.9)		
Learning attitude		50.1 ± 5.5	55.0 ± 5.1	52.5 ± 5.6	4.62	0.001
Problem-solving ability		149.1 ± 15.0	157.1 ± 4.8	152.7 ± 15.0	2.72	0.008
Empathetic ability		41.4 ± 4.0	41.3 ± 4.1	41.2 ± 4.0	−0.05	0.961

**Table 2 healthcare-10-00644-t002:** Effect of the programs between the two groups.

Variables	Group	Pre-Test	Post-Test	Difference	t	*p*	F	*p*
		M ± SD	M ± SD	M ± SD				
Learning attitude	Exp.	50.1 ± 5.5	54.5 ± 4.8	4.3 ± 4.4 1.4 ± 6.9	−2.555	0.012	0.486	0.487
	Cont.	55.0 ± 5.1	56.2 ± 6.9					
Problem solving	Exp.	149.1 ± 15.0	154.0 ± 15.3	4.7 ± 10.9				
	Cont.	157.1 ± 4.8	153.7 ± 16.0	−2.7 ± 14.4	−2.917	0.004	4.012	0.048
Empathy	Exp.	41.4 ± 4.0	43.4 ± 5.7	1.9 ± 6.7	−2.014	0.047	-	-
	Cont.	41.3 ± 4.1	40.8 ± 4.0	−0.5 ± 5.6				

## Data Availability

Data are available upon substantiated request from the corresponding author.

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
