# Peer review of "Effects of Situation-Based Flipped Learning and Gamification as Combined Methodologies in Psychiatric Nursing Education: A Quasi-Experimental Study"

_healthcare, 2022, doi:10.3390/healthcare10040644_

Round 1
Reviewer 1 Report
I find that this paper presents an interesting approach to blended learning methods. However I would recommend that for the interests of the readers more information and detail are given for the gamification elements as well as flipped learning examples.
Although the overall framework of the design is represented diagrammatically, the readers may still feel puzzled as to the extent of the gamification during lectures, the duration of such activities, examples of activities as well as examples of flipped learning.
It is also rather unclear whether the participants who did not volunteer for the study followed the same program.
I also find there is a lack of information about how the authors designed a gamified environment to support an increase in learners' empathy levels. There can be more detail added in the Methods section.
Author Response
Thank you for your opportunity to revise our manuscript accordingly. We appreciate this careful review and constructive suggestions. The manuscript has been substantially improved after the suggested edits were made.

Reviewer 2 Report
Dear Authors,
Thank you for the opportunity to read your manuscript.
Effective and efficient teaching in nursing is a big challenge. Teaching students appropriate behaviours and attitudes especially in challenging situations is an important task.
On the one hand, we have had studentes living in a technological world almost from birth. They ability to focus attention is definitely less in the past. On the other hand, teaching methods, new technologies and their use in education mean that today's classes are nothing like those of years ago. All this forces a change in the approach to teaching.
Research shows that the use of active methods such as simulation, flipped classroom and gamification are used with great success in preparing for medical professions. Your results support this thesis.
Despite the interesting presentation of the topic I have a few suggestions:
1 - the purpose of the work is missing
2 - it would be good to present the topices of the class
3 - give examples of games
I think it would be useful to explain to your readers how you conducted the activities.
I congratulate you on your idea and implementation. I hope that you will use these methods with the next generation of students.
Author Response

(The authors gave the same response as above.)
